

# Identification and verification of prognostic genes related to zinc homeostasis and zinc transport in breast cancer

Mengxuan Li[1,*], Haoyi Zi[1,2,*], Jiajun Ding[1], Shuai Wang[1], Yujie Bai[1,3], Jianing Sun[1,3], Cong Fan[1], He Chen[1] and Ting Wang[1]

[1] Department of Thyroid, Breast and Vascular Surgery, Xijing Hospital of the Fourth Military Medical University, Xi'an, Shaanxi, China
[2] Shaanxi University of Chinese Medicine, Xianyang, Shaanxi, China
[3] Xi'an Medical University, Xi'an, Shaanxi, China
[*] These authors contributed equally to this work.

## ABSTRACT

**Purpose.** Zinc homeostasis and zinc transporter (ZHT) have been closely associated with the development of various cancers. Therefore, in this study, prognostic genes and their mechanisms related to ZHT in breast cancer (BC) were explored.
**Patients and methods.** Differential expression analysis and weighted gene co-expression network analysis (WGCNA) were utilized to identify genes associated with Zinc homeostasis and Zinc transporter-related genes (ZHTGs) in BC. Subsequently, independent prognostic factors and their correlations with clinical features were examined to investigate their association with the prognosis of BC. Finally, we further explored the pathways and immune cells associated with BC prognosis. We also verified gene expression in tissues and cells by quantitative polymerase chain reaction (qPCR).
**Results.** In this study, six prognostic genes were identified. Patients were subsequently classified into high-risk and low-risk cohorts based on the median risk score, with the low-risk group presenting superior survival outcomes. Subsequently, riskScore, age, tumor/node/metastasis (T/N/M) stage showed significant associations with the prognosis of BC, and the constructed nomogram demonstrated strong predictive performance. Clinical analysis revealed differences in risk scores among sub-cohorts with different clinical characteristics, such as race (white and others) and T-stage (T1 and T2, T1 and T3). Furthermore, significant disparities were noted in immune cells and immune checkpoints across different risk cohorts. The results of reverse transcription quantitative PCR were basically consistent with the prediction. In addition, the IHC results from the Human Protein Atlas database further validated our prediction.
**Conclusion.** We screened six prognosis genes related to ZHT in BC, providing a reference for the prognosis and personalized treatment of BC.

Corresponding author
Ting Wang, ting_w100@126.com

## INTRODUCTION

Breast cancer (BC) is the most common type of cancer in women worldwide (11.6% of all cancers globally) and the leading cause of cancer-related death in women (*Bray et al., 2024*). BC can be divided into four major subtypes based on the expression of molecular markers (estrogen receptor (ER), progesterone receptor (PR) and human epidermal growth factor 2 (HER2)) (*Waks & Winer, 2019*). However, with standard clinical diagnosis and treatment, approximately one in five to one in three BC patients will develop distant metastases, which are the leading cause of death in most (approximately 90%) breast cancer patients (*Britt, Cuzick & Phillips, 2020*; *Jabbarzadeh Kaboli et al., 2020*). In addition, BC is highly heterogeneous, which somewhat limits the broad applicability of classification and standard care (*Waks & Winer, 2019*). Therefore, it is important to further explore the relevant characteristics of BC and potential new therapeutic targets.

Zinc is an essential trace element required for gene regulation, enzyme activity, and protein structure. It also plays a crucial role as a cofactor in the metabolism and cellular processes of over 300 enzymes in the human body. Zinc plays a crucial role in maintaining the structural stability and DNA-binding activity of approximately 2,000 transcription factors *in vivo* (*Chasapis et al., 2012*; *Skrajnowska & Bobrowska-Korczak, 2019*). In addition, zinc as a second messenger is widely involved in important biological processes such as cell proliferation and differentiation, cell cycle regulation, and cell apoptosis (*Sharif et al., 2012*; *Yamasaki et al., 2007*). Zinc transporters are the key molecules regulating the homeostasis of zinc ions in cells and are involved in various physiological processes by mediating the transmembrane transport of zinc ions (*Kambe et al., 2015*). Its family members such as ZIPs and ZnTs are respectively responsible for the intake and excretion of zinc ions, and play important roles in growth and development, signal transduction and gene expression (*Myers et al., 2017*; *Yin et al., 2023*). Studies have shown that abnormal zinc transporters are associated with various diseases. For instance, ZIP13 promotes the metastasis of human ovarian cancer cells by activating the steroid receptor coactivator/focal adhesion kinase (Src/FAK) signaling pathway (*Cheng et al., 2021*), while ZnT1 is related to the prognosis of patients with hepatocellular carcinoma (*Kakita et al., 2024*). In addition, zinc transporters also play an important role in immune regulation by regulating zinc homeostasis (*Dwivedi et al., 2019*; *Wessels, Maywald & Rink, 2017*). However, the prognostic role of zinc homeostasis and zinc transporter-related genes (ZHTGs) in BC and their impact on the tumor microenvironment are not yet clear and require further investigation.

Therefore, this study conducted bioinformatics analysis based on the BC transcriptome dataset in the public database and ZHTGs. The aim is to identify the prognostic genes related to ZHT in BC. Meanwhile, the prognostic model was constructed based on the identified prognostic genes. In addition, the immune microenvironment in BC was also explored. It provides a new direction for discovering new immunotherapy and targeted therapy strategies for BC.

## MATERIALS & METHODS

### Data extraction

The Cancer Genome Atlas Breast Invasive Carcinoma (TCGA-BRCA) dataset, including survival information, age, and tumor node metastasis (TNM) staging, was downloaded from The Cancer Genome Atlas (TCGA) database (http://cancergenome.nih.gov/). Patients without survival information were excluded, resulting in tissue from 1,082 BC and 113 control specimens (*Li et al., 2022*). The GSE20685 dataset was obtained from the Gene Expression Omnibus (GEO) database (https://www.ncbi.nlm.nih.gov/geo/). It comprises 327 BC tumor tissue specimens with complete survival information, sequenced on the GPL570 platform (*Luo et al., 2022*). A search was conducted in Molecular Signatures Database (MSigDB, https://www.gsea-msigdb.org/) for the keywords "WP_ZINC_HOMEOSTASIS" (Human Gene Set, WP3529, 37 genes) and "GOBP_ZINC_ION_TRANSPORT" (Human Gene Set, GO:0006829, 28 genes) to retrieve ZHTGs. After removing duplicates, a total of 40 ZHTGs were retained (Table S1).

### Differential expression analysis

The TCGA-BRCA dataset underwent analysis for differential gene expression employing the DESeq2 package (v 1.34.0) (*Love, Huber & Anders, 2014*) to evaluate the differentially expressed genes (DEGs) between BC and control specimens (p.adjust <0.05 and |log2FoldChange (FC)|>1).

### Weighted gene co-expression network analysis

Weighted gene co-expression network analysis (WGCNA) is a bioinformatics approach based on gene expression correlation to construct modular networks. It clusters genes with similar expression patterns, analyzes the associations between modules and specific traits/phenotypes, and identifies key functional gene groups (*Langfelder & Horvath, 2008*). In this study, WGCNA was applied to analyze BC transcriptome data to identify co-expression modules associated with ZHTGs, thereby screening their related genes. First, univariate Cox regression analysis based on the expressions of 40 ZHTGs in the TCGA-BC dataset was performed using the survival package (v 3.5-3) (*Lei et al., 2023*) (Hazard Ratio (HR) $\neq$ 1 & $p$ value < 0.05) to screen ZHTGs linked to BC prognosis for subsequent analysis. Following this, utilizing the ZHTGs, single-sample Gene Set Enrichment Analysis (ssGSEA) was employed to compute ZHTGs scores for BC specimens. These BC specimens were subsequently segregated into high and low score groups using the ideal threshold value of the scores. The survminer package (v 0.4.9) (*Liu et al., 2021*) was used to plot Kaplan–Meier (K–M) survival curves among high/low score cohorts ($p$ value < 0.05). Next, the WGCNA package (v 1.70.3) (*Langfelder & Horvath, 2008*) was employed to construct a co-expression network using ZHTGs scores as traits, aiming to identify module genes most correlated with the traits. Particularly, all BC samples were grouped to detect and eliminate anomalies. A soft threshold ($\beta$) with a connectivity close to 0 and an R2 value great than 0.85 was selected. A scale-free network was built using selected soft threshold, and a hybrid dynamic tree cutting algorithm with a cutting tree parameter of 0.4 (the minimum gene number per module is 100, and the module merging

parameter is 0.25) was used to identify the co-expression modules. Subsequently, the correlation coefficients between these modules and the traits were calculated, and the top two modules with the strongest positive and negative correlations were selected as key modules. Finally, by setting the gene significance (GS) to 0.3 and module membership (MM) to 0.3, key module genes were filtered and retained to select more important key module genes.

### Enrichment analyses

The obtained DEGs were intersected with more important key module genes to obtain a set of candidate genes. To investigate the function in which the candidate genes were involved, Gene Ontology (GO, p.adjust < 0.05) and Kyoto Encyclopedia of Genes and Genomes ($p$ value < 0.05, KEGG) enrichment analyses were performed using the clusterProfiler package (v 4.2.2) (*Wu et al., 2021*).

### Development and verification of risk models

Within the TCGA-BRCA dataset, candidate genes were subjected to univariate Cox regression analysis *via* the survival package (*Lei et al., 2023*) to identify genes associated with BC prognosis, with HR $\neq$ 1 and $p$ value < 0.01 as the screening criteria. Subsequently, a least absolute shrinkage and selection operator (LASSO) regression analysis was executed using the glmnet package (v 4.1-4) (*Sasikumar et al., 2022*) grounded on the findings of the previous step to further refine prognostic genes. The risk score for each individual was computed using the subsequent equation:

$$\text{risk score} = \sum_{i=1}^{n} \text{coef(genei)} \star \text{expr(genei)}.$$

Individuals were subsequently classified into high/low risk cohort according to the median risk score. Subsequent Kaplan–Meier (K-M) survival analysis using the survminer package (*Liu et al., 2021*) to plot K-M curves ($p$-value < 0.05). The survival receiver operating characteristic (ROC) package (v 1.0.3) (*Heagerty, Lumley & Pepe, 2000*) was utilized to generate ROC curves for 1-, 3-, and 5-year periods to assess the prognostic performance of BC patients. The external dataset GSE20685 for BC patients was employed to validate the constructed risk model in this study.

### Recognition of independent prognostic factors

To further investigate the prognostic implications of clinical pathological factors such as Age, Race, *T/N/M* stage were included along with the risk score in the prognostic model for univariate Cox regression analysis (HR $\neq$ 1, $p$ value < 0.05) in BC specimens from the TCGA-BRCA dataset. Subsequently, a proportional hazards (PH) assumption test was conducted to select genes ($p$ value > 0.05). Factors fulfilling the PH presumption were subsequently incorporated in multivariate Cox regression analysis, where factors with a $p$ value < 0.05 were defined as independent prognostic factors.

### Construction of nomogram

In order to further investigate the prognostic implications of independent prognostic factors, a nomogram forecasting the longevity likelihood of BC specimens was constructed

using the rms package (v 6.5-1) (*Sachs, 2017*). The anticipatory effectiveness of this model was assessed *via* calibration curves and ROC.

### Correlation analysis of clinical characteristics
In the TCGA-BRCA dataset, rank sum examinations were used to scrutinize the variations in risk scores among subgroups with diverse clinical characteristics and the survival disparities among two risk groups in the subgroups with diverse clinical characteristics.

### Gene set enrichment analysis
In the TCGA-BRCA dataset, based on the two risk cohorts, differential analysis was carried out *via* DESeq2 to calculate log2 fold change (log2FC). The log2FC values were then sorted from largest to smallest, followed by gene set enrichment analysis (GSEA) were carried out *via* clusterProfiler package (*Yu et al., 2012*), using the h.all.v2023.1.Hs.symbols. gmt from MisgDB database as the background gene set (FDR < 0.05).

### Gene set variation analysis
To delve deeper into the variations in KEGG pathways among two risk cohorts in TCGA-BRCA, gene set variation analysis (GSVA) was executed utilizing the GSVA package (v 1.42.0) (*Hänzelmann, Castelo & Guinney, 2013*) based on the background gene set "c2.cp.kegg.v2023.1.Hs.symbols.gmt" to obtain enrichment scores for different pathways. The limma package (v 3.54.0) (*Love, Huber & Anders, 2014*) was then utilized to compare the functional enrichment pathways among two risk cohorts, with a threshold of $p$ value < 0.05 to select key pathways.

### Immune-related analysis
In the TCGA-BRCA dataset, the enrichment scores of 28 immune-infiltrating cells for all samples among risk cohorts were computed using the ssGSEA technique. Subsequently, differential immune infiltrating cells were compared among different risk cohorts. Using the psych package (v 2.1.6) (*Revelle, 2021*), the Spearman correlation between differential immune infiltrating cells and prognosis genes was calculated (|cor| > 0.3 & $p$-value < 0.05). In the TCGA-BRCA dataset, we contrasted the levels of eight immune checkpoint molecules (CD274, LAG3, CTLA4, TIMP3, PDCD1, PDCD1LG2, TJAP1, LGALS9) among two risk cohorts. Additionally, the Tumor Immune Dysfunction and Exclusion (TIDE) online database (http://tide.dfci.harvard.edu/) was employed to acquire TIDE score, dysfunction score, and exclusion score for the specimens in the TCGA-BRCA dataset, and inter-cohort differences were compared. TIDE score is a computational method for evaluating tumor immune microenvironment functionality, which reflects the degree of tumor immune evasion. The dysfunction score is an indicator derived from the expression patterns of T cell exhaustion-related genes, assessing whether tumor-infiltrating T cells lose effector functions due to chronic antigen stimulation. The exclusion score evaluates whether a tumor forms an "immune-excluded" microenvironment by analyzing genes related to tumor stromal fibrosis, abnormal angiogenesis, and other factors that inhibit immune cell infiltration. To evaluate the ratio of immune therapy response in two cohorts, chi-square test was executed to examine the variations in immune reaction.

## Cell culture and tissue samples collection

MCF-10A, T47D, BT474, MDA-MB-231, SUM-149, SUM-159 and MCF-7 were purchased from American Type Culture Collection. Due to the different molecular types of breast cancer and the existence of heterogeneity, we selected immortalized normal epithelial cells MCF-10A as the control cell line; According to the different expression statuses of ER, PR and HER-2, six breast cancer cell lines, namely T47D, BT474, MDA-MB-231, SUM-149, SUM-159 and MCF-7, were selected as the experimental groups to analyze the expression of related genes in the breast cancer cell lines. MCF-10 A cells were grown in DMEM/F12 medium supplemented with 5% horse serum and growth supplements (Zhong Qiao Xin Zhou Biotechnology, Shanghai, China). T47D, BT474, MDA-MB-231, SUM-149, SUM-159 and MCF-7 cells were cultured in DMEM (Gibco, USA) supplemented with 10% fetal bovine serum (FBS; Gibco, Waltham, MA, USA). All cell lines were cultured in 5% $CO_2$ at 37 °C in a humidified atmosphere. We obtained 10 pairs of BC tissue and adjacent normal tissue from patients without preoperative chemotherapy, endocrine therapy, or radiotherapy who had undergone tumor resection at The Xijing Hospital of the Fourth Military Medical University. The study was approved by the hospital's ethics committee (KY20232266-C-1), and the content of the study received written informed consent from patients. All methods were performed in accordance with the relevant guidelines and regulations.

## RNA isolation and reverse transcription quantitative PCR (RT-qPCR)

Total RNA was extracted from cells or tissues using SPARKeasy Improved Tissue/Cell RNA Kit (Sparkjade, Shandong Sparkjade Biotechnology Co., Ltd., Shandong, China) following the manufacturer's instructions. cDNA was synthesized using SPARKscript II All-in-one RT SuperMix for qPCR (Sparkjade). The reverse transcriptional reaction program lasted 15 min at 50 °C and 5 s at 80 °C. We performed qPCR using the Quant Studio 7 Pro (Applied Biosystems). RT-qPCR was performed using 2×SYBR Green qPCR Mix (Sparkjade). RT-qPCR reaction program: preincubation at 94 °C for 3 min, 40 cycles of amplification with 10 s at 94 °C, 20 s at 55—60 °C, followed by an extension at 72 °C for 30 s. The internal controls were $\beta$-actin. Gene expression levels were quantitatively calculated by the 2-$\Delta\Delta$Ct method. Table S2 provides sequences of primers used in this research.

## Immunohistochemistry

The Human Protein Atlas (HPA) (https://www.proteinatlas.org/) (*Uhlen et al., 2017*) contains IHC profiles of normal and tumor tissues.

## Statistical analysis

In R software (v 4.1.0; *R Core Team, 2021*), the data was processed and analyzed. Differences among cohorts were assessed using the Wilcoxon rank-sum test or chi-square test, with a significance threshold of $p$ value < 0.05 indicating statistical significance. Continuous variable data were analyzed using Student's $t$-test or Wilcoxon test, and categorical data were analyzed using chi-square test. Survival differences were compared

using Kaplan–Meier analysis and log-rank test. *$P < 0.05$, **$P < 0.01$, ***$P < 0.001$, ****$P < 0.0001$.

## RESULTS

### ZHTGs were associated with survival of BC

In the TCGA-BRCA dataset, there were 4,948 DEGs, consisting of 2,986 genes that were up-regulated and 1,962 genes that were down-regulated (Fig. 1A). Through univariate Cox regression analysis, three ZHTGs associated with BC prognosis were selected, with SLC30A5 identified as a high-risk gene (HR > 1), while SLC1A1 and TMEM163 as low-risk genes (HR < 1) (Fig. 1B). Subsequently, based on the optimal cutoff value of three genes scores, the BC specimens in the TCGA-BRCA dataset were split into two score cohorts. It was found that patients in the high ZHTGs score cohort had significantly higher survival rates than those in the low ZHTGs score cohort ($p$ value = 0.01), suggesting that ZHTGs may impact the outcome of BC patients (Fig. 1C). Then, the ZHTGs score was used as a trait to further screen its related genes through WGCNA. Initially, no obvious outlier specimens were observed in the dataset, hence no specimens needed to be excluded (Fig. S1A). By setting the $\beta$ value to 6, we achieved an $R^2$ approaching 0.85 and a connectivity close to 0 (Fig. 1D). Then, 16 co-expression modules were identified using the mixed dynamic tree-cutting algorithm, excluding the grey module (Fig. S1B). Correlation analysis revealed that the brown module ($R = 0.41$) and the black module ($R = -0.36$) had the strongest correlations with trait, with the brown module containing 1,496 genes and the black module containing 766 genes (Fig. 1E). Finally, by setting the GS and MM thresholds to 0.3, a total of 443 more important key module genes were obtained, with 313 genes retained in the brown module and 130 genes retained in the black module (Figs. S1C–S1D).

### There were six prognosis genes were screened in BC

There were 134 candidate genes were acquired by intersecting 4,948 DEGs with 443 key module genes (Fig. 2A). Subsequently, these candidate genes were further explored for potential enrichment of functions through GO and KEGG analyses. In the GO analysis, these candidate genes were found to be primarily associated with cell cycle and chromosome-related functions, such as mitotic cell cycle phase transition and chromosomal region (Fig. 2B). In the KEGG term, the candidate genes were also involved in pathways related to cell cycle, prostate cancer, and BC (Fig. 2C). Then, a total of six prognosis-related genes (CLIC6, EIF4E3, TFF1, TPRG1, RSPH1, PCSK6) were identified through univariate Cox regression analysis ($p$-value < 0.01 & HR < 1) (Fig. 2D). Furthermore, these six genes were further confirmed as prognostic genes using the LASSO algorithm (Fig. 2E).

### The risk model in TCGA-BRCA and GSE20685 datasets has strong predictive performance

Based on the median of the risk score (Risk score = CLIC6 * (−0.0591) + EIF4E3 * (−0.1226) + TFF1 * (−0.0145) + TPRG1 * (−0.017) + RSPH1 * (−0.0272) + PCSK6 *

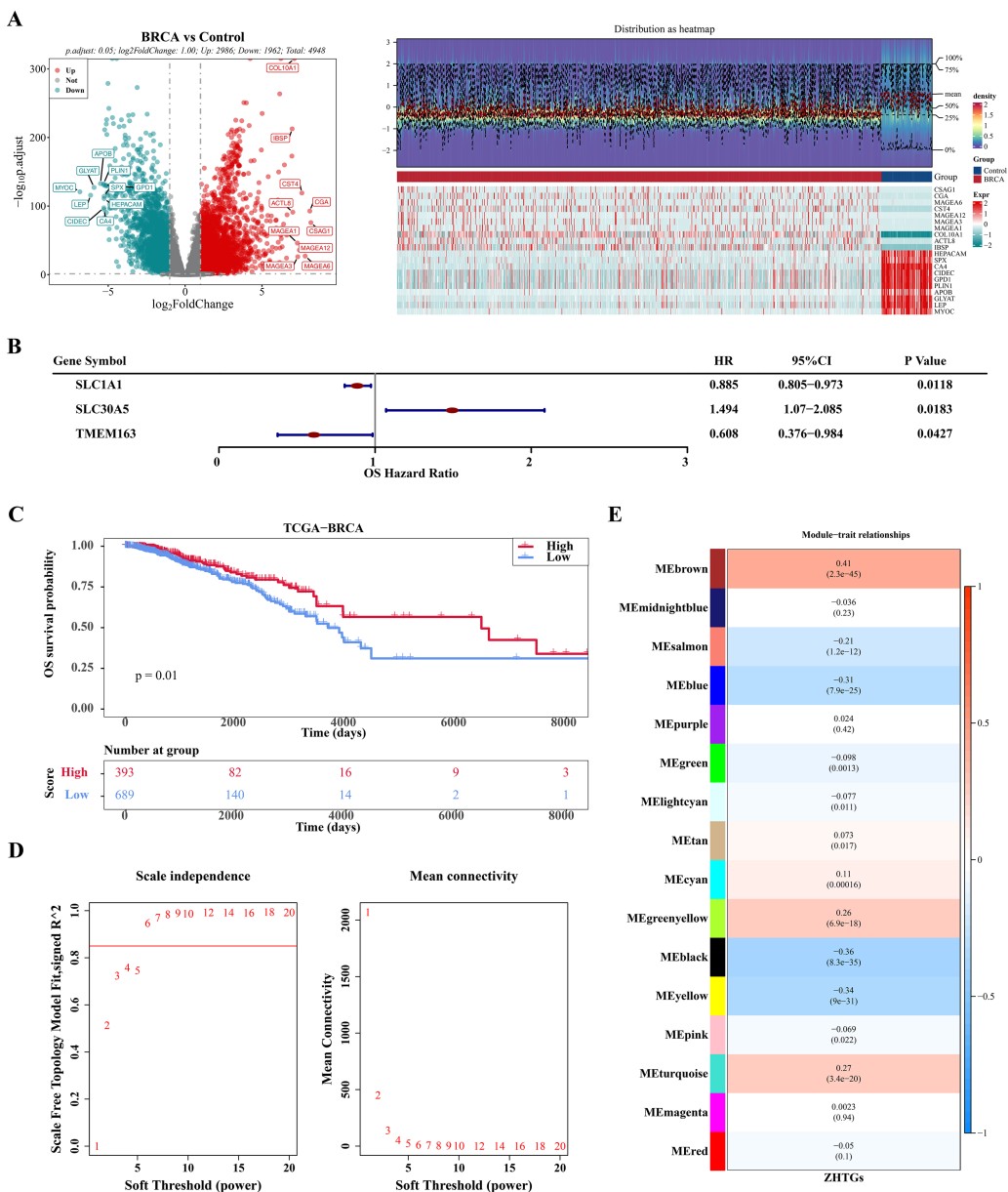

**Figure 1 WGCNA screening for ZHTGs in BC.** (A) The volcano plot and heat map clearly shows the DEGs, with a total of 4,948 DEGs. The top 10 differential genes were down-regulated according to log2FC sequencing. (B) Forest plot of univariate Cox regression analysis shows the three ZHTGs significantly associated with overall survival (OS) in breast cancer patients. (C) K–M survival curves shows patients in the high ZHTGs score cohort had significantly higher survival rates than those in the low ZHTGs score cohort. (D) The optimal soft threshold 6 is used to construct the co-expression network. (E) Associations between phenotypes and modules were constructed using ZHTGs scores of BC samples.

(−0.0796)), BC specimen of TCGA-BRCA dataset were stratified into two risk cohorts. It was noticed that as the risk scores rose in the BC specimens, the number of deaths notably increased (Fig. 3A). Subsequently, from the K-M curves, it was evident that the survival rate of individuals in the high-risk cohort was lesser in comparison to those in

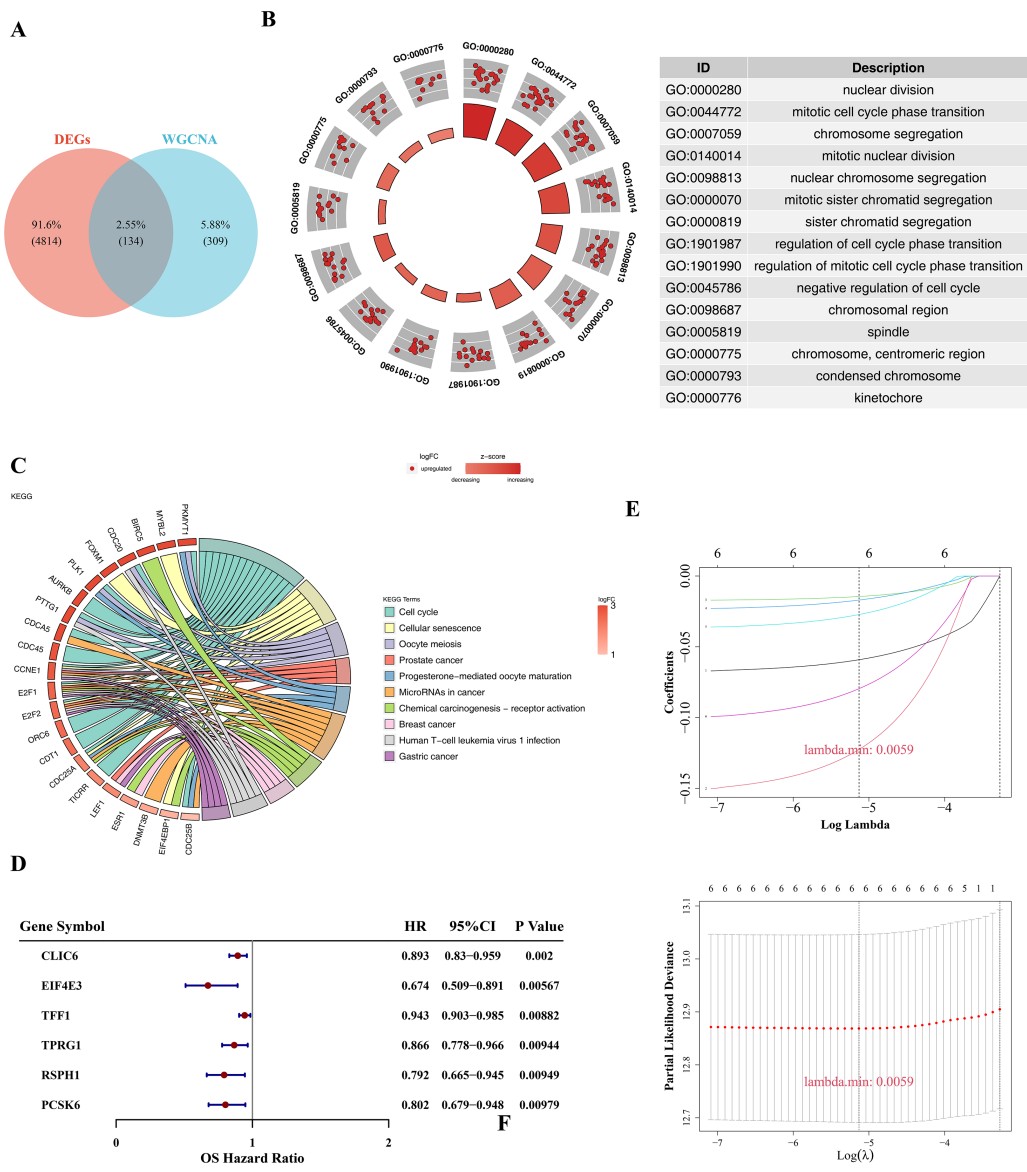

**Figure 2** **Prognostic gene screening.** (A) The Venn diagram showed that a total of 134 candidate genes were obtained by crossing 4,948 DEGs with 443 key module genes. (B) GO enrichment analysis was used to explore potential functional enrichment of candidate genes, showing significant top 15 pathways. The left inner circle is a bar chart, the height of the bar chart represents the significance of the Term, the higher the higher the significance; The color of the bar represents the *z*-score, and the darker the color, the larger the *z*-score. The *z*-score is not a standard statistic, but can indicate whether a biological function is more likely to be up-regulated or down-regulated. The outer circle shows a scatter plot of the expression levels of each gene in each Term, with red and blue representing up-regulated and down-regulated genes, respectively. On the right is the GO enrichment entry description. (C) KEGG enrichment analysis revealed significant top 10 pathways, with the color of the gene band on the left representing log2FC of the gene, and different color bands on the right representing different pathways. (D) Forest plot of univariate Cox regression analysis shows six prognosis-related genes. (E) The LASSO coefficient profile of six prognostic-related genes and the tenfold cross-validation for variable selection in the LASSO model.

the low-risk cohort (Fig. 3B). Furthermore, ROC curves were plotted at 1, 3, and 5 years as survival time nodes, demonstrating that the AUC values at all three time points were higher than 0.6, suggesting good forecasting ability of the model (Fig. 3C). Finally, the universality of the risk model was further validated in the GSE20685 dataset. The results indicated that similar to TCGA-BRCA, this risk model of this dataset could effectively predict patient survival (Figs. 3D–3F). Additionally, the GSE20685 dataset showed more distinct separation of survival groups than the TCGA-BRCA dataset, which further supported the validity of the model.

## There were five independent prognostic factors

Within the univariate Cox regression analysis, the *p*-values of risk score, age, *T/N/M* stage were all less than 0.05 and passed the PH assumption (Fig. 4A). Subsequently, riskScore, age, *T/N/M* stage were screened as independent prognostic factors *via* multivariate Cox regression analysis (Fig. 4B). To extend the analysis of survival prediction in BC patients by these factors, a nomogram was developed (Fig. 4C). Calibration curve results indicated that the slope of the curve closely approximated the diagonal line, suggesting high prediction accuracy (Fig. 4D). Moreover, with AUC values consistently exceeding 0.7, this suggested that the model possessed robust predictive capability (Fig. 4E).

## Clinical feature was associated with risk score and survival rate

By comparing the differences in risk scores between different clinical feature sub-cohorts to explore their relationships, significant differences were observed in the risk scores between the race (white and others) and T stage (T1 and T2, T1 and T3) sub-cohorts (Fig. 5A). Subsequently, the K-M survival differences between the high- and low-risk groups under different clinical characteristics were compared. The results revealed significant survival differences between the high-risk and low-risk groups across various clinicopathological characteristics, including age ≤60, T stage 1–2, N stage 1–3, and M stage 0. Notably, the survival rate of the high-risk group was consistently lower across all these characteristics (Fig. 5B).

## The pathways with differences between the high- and low-risk groups

To explore the significantly enriched pathways between the high- and low-risk groups, GSEA and GSVA analyses were performed. The results revealed that DEGs among two risk cohorts were mainly involved in E2F TARGETS, G2M CHECKPOINT, and KRAS SIGNALING DN pathways (Fig. 6A). Subsequently, to further explore the differences in KEGG pathways between the high- and low-risk groups, GSVA was conducted. The results showed that there were 29 differential KEGG pathways between the two cohorts, including JAK-STAT signaling pathway, *β*-alanine metabolism, ribosome, drug metabolism-cytochrome P450, *etc.* (Figs. 6B–6C).

## Patients in different risk cohorts had different effects on the immune response

For additional investigation into the variances in the immune microenvironment among two risk cohorts, a series of immune-linked analyses were performed. A heatmap was

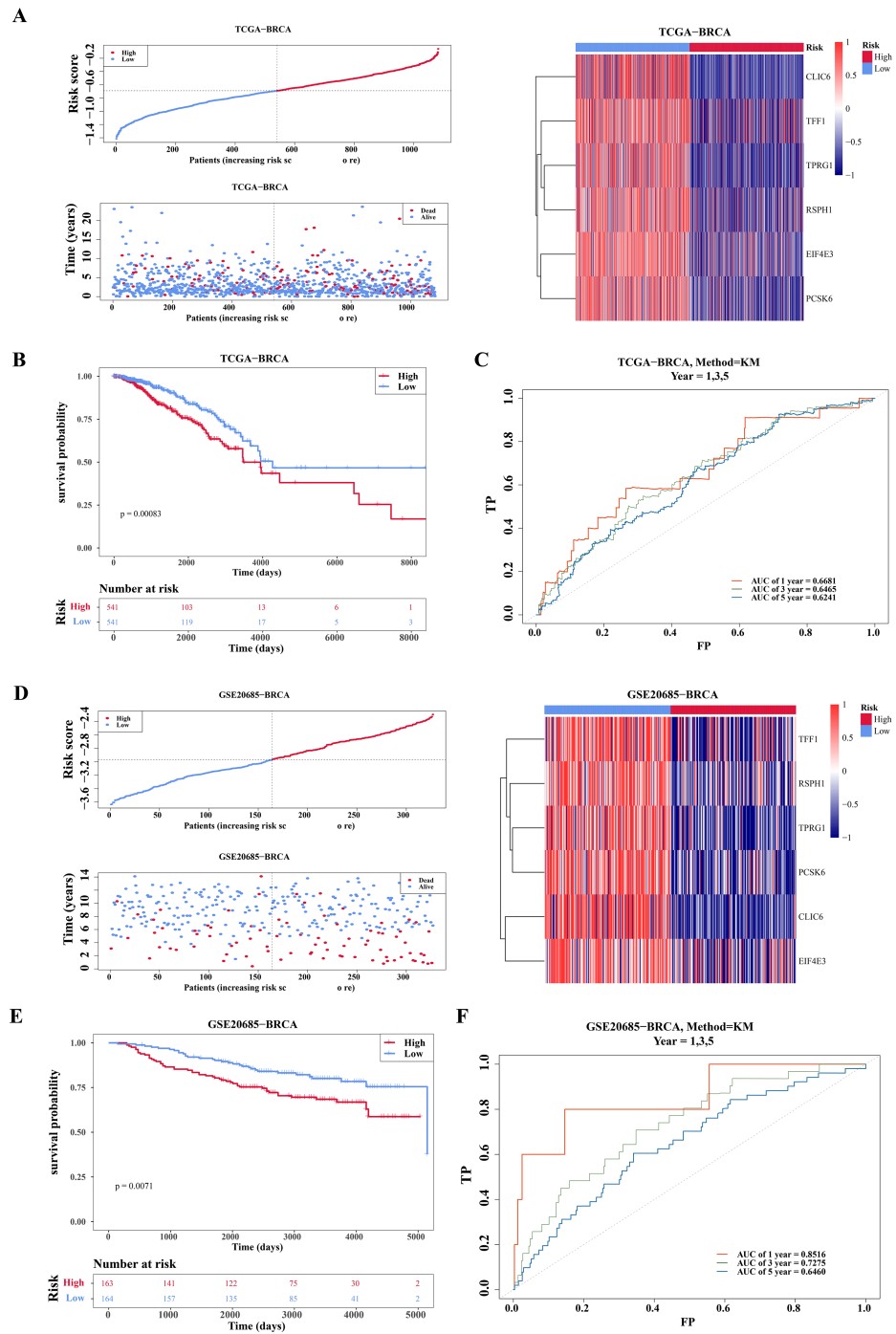

**Figure 3 Evaluation and validation of prognostic model.** (A) Risk scores, survival status, and heat map of gene expression of prognostic genes between high and low risk groups in the training set (TCGA-BRCA). (B) K–M survival curves of OS for patients between low-risk and high-risk groups in the training set. Red represents the high-risk group and blue represents the low-risk group. The *x*-axis represents time in days, while the *y*-axis shows the OS survival probability. (C) ROC curves for predictive performance of

employed to exhibit the spread of 28 immune infiltrating cell enrichment scores among two cohorts in the TCGA-BRCA dataset (Fig. 7A). Following this, differential analysis uncovered substantial differences in the enrichment scores of 24 immune cells among the two risk cohorts. Specifically, central memory CD8 T cell, eosinophil, mature dendritic cell, mast cell, memory B cell, natural killer cells (NK cells), neutrophil, plasmacytoid dendritic cell had higher scores in the low-risk cohort, while other cells were opposite (Fig. 7B). Further correlation analysis revealed significant correlations between six prognostic genes and various differentially expressed immune infiltrating cells. Among them, the strongest negative correlation was observed among RSPH1 and activated CD4 T cell (cor = −0.46) (Fig. 7C, Tables S3–S4). Additionally, we found obvious variances in the expression of seven immune checkpoints (LAG3, CTLA4, TIMP3, PDCD1, PDCD1LG2, TJAP1, LGALS9) between high and low-risk cohorts, with all six immune checkpoints, except TIMP3, displaying elevated expression in the high-risk cohort (Fig. 7D). Furthermore, significant differences were observed in TIDE score and Dysfunction score among two cohorts, with higher scores in the low-risk cohort (Fig. 7E). Furthermore, notable variations were observed in immune treatment responses among two cohorts (Fig. 7F). The results indicated that there might be implications for the effectiveness of immunotherapy in different risk cohorts.

**External and experimental validation of the six model related genes**

Compared with the paired adjacent tissues, the expression levels of CLIC6, TFF1, TPRG1, RSPH1 and PCSK6 were up-regulated. The expression levels of EIF4E3 were down-regulated in BC specimens (Figs. 8A–8F). In addition, we also investigated the expression of prognostic genes in BC cell lines (including T47D, BT474, MDA-MB-231, SUM-149, SUM-159, and MCF-7), using MCF-10A as the control cell line. Compared with the MCF10A cell line, the gene expression level in most cell lines was overall consistent with the tissue verification results (Figs. 9A–9F). Finally, in order to verify the expression of the six genes involved in building the prognosis model, we downloaded immunohistochemical staining images from the HPA database. CLIC6, TFF1, TPRG1, RSPH1 and PCSK6 were expressed at notably different levels between BC and normal breast tissues. There was no notable difference in the expression of EIF4E3 (Figs. 10A–10F). This may be due to the low consistency between antibody staining and RNA expression data. These results supported our hypothesis and provided evidence for the rationality of selecting these six genes to build the prognosis model.

## DISCUSSION

BC is closely associated with ZHT. Studies have shown that zinc plays an important role in the occurrence and development of BC, closely related to key biological processes such

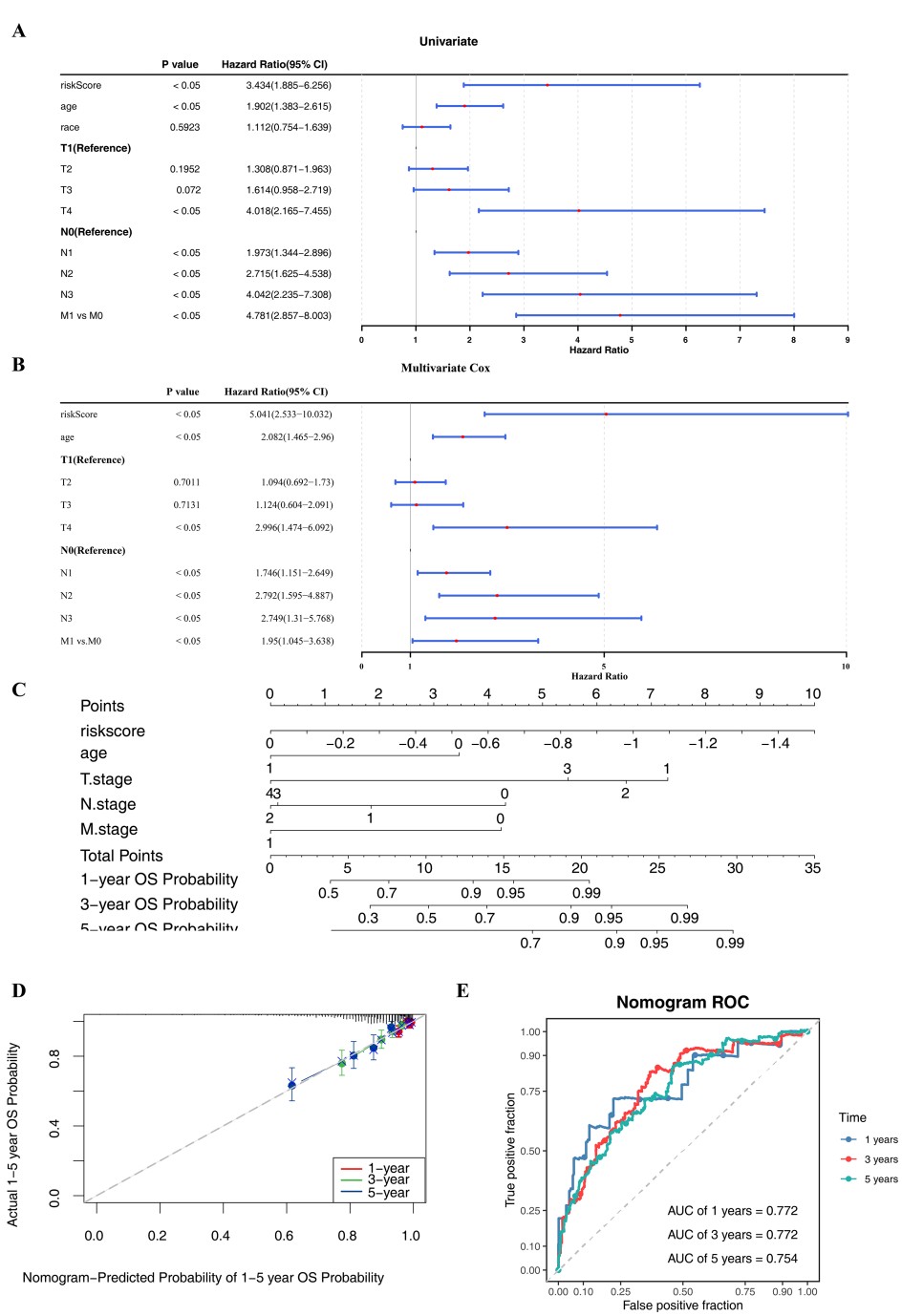

**Figure 4  Independent prognostic analysis and nomogram construction.** (A–B) Univariate and multi-variate Cox analyses of clinical factors and risk score with OS in training set. Risk Score, age, T/N/M.stage were screened as independent prognostic factors. (C) Nomogram predicting 1, 3 and 5-year survival rate of breast cancer patients. Each independent prognostic factor was assigned a corresponding score, and the total score was obtained by summing the scores of all factors. The total score was then used to predict the 1-, 3-, and 5-year OS probabilities of breast cancer patients. Higher scores were associated with higher OS 

**Figure 4 (…continued)**
probabilities. (D) The calibration curves for 1, 3 and 5-year OS in training set. The slopes of the calibration curves were all close to 1, indicating that the nomogram had a good predictive performance. (E) ROC curves for predictive performance of the nomogram model in training set.

as cell proliferation, metastasis, and apoptosis (*Qu et al., 2023*). Regulating the levels of zinc inside and outside BC cells may affect tumor growth and prognosis (*Takatani-Nakase, 2018*). The recurrence and metastasis of BC are currently the biggest challenges encountered in the treatment process. Therefore, further exploration of the relationship between BC and zinc homeostasis, zinc transport, and their regulatory mechanisms is of great significance. This study found that SLC30A5 might be a risk factor for BC. SLC30A5, also known as ZnT5, is a member of the zinc transporter SLC30 family (*Liu et al., 2024b*). The zinc transporters of the SLC30A family are mainly responsible for transporting zinc ions out of the cytoplasm or transferring them to intracellular organelles, thereby regulating intracellular zinc homeostasis. Studies have shown that the expression patterns of the SLC30A family genes in gastric cancer are diverse and may have different prognostic significance (*Guo & He, 2020*). In addition, SLC30A5 and ZnT6 involved in the formation of dimers, may affect the BC of epithelial mesenchymal cell transformation (EMT), will affect the progress of the BC. In conclusion, SLC30A5 may affect the occurrence and development of BC by regulating zinc homeostasis and the EMT process, and may become a potential therapeutic target. The prognostic model established in this study mainly includes the following six prognostic genes: CLIC6, EIF4E3, TFF1, TPRG1, RSPH1 and PCSK6. Chloride intracellular channel 6 (CLIC6) is one of the family members of chloride intracellular channels. Microarray studies have identified changes in CLIC6 expression in BC tissues. Eukaryotic translation initiation factor 4E family member 3 (EIF4E3) is a member of the EIF4E family of proteins that bind to the 5'-cap structure of messenger RNA (*Osborne et al., 2013*). Studies have shown that high expression of EIF4E3 gene is more conducive to patient survival (*Li et al., 2024*). Trefoil Factor 1 (TFF1) is an estrogen-inducible protein, expressed in BC and some digestive tumors. The regulation of tumor protein P63 regulated 1 (TPRG1) in tumor tissues is closely related to early tumor recurrence (*Hong et al., 2022*). Radial spoke head component 1 (RSPH1) encodes an acidic protein associated with chromosomes during the metaphase of male meiosis. BC patients with high expression of RSPH1 gene have a better prognosis (*Yu, He & Xu, 2022*). Proprotein convertase subtilisin/kexin type 6 (PCSK6) is a serine protease, and patients with high expression of PCSK6 have a longer overall survival (OS) (*Sethi et al., 2023*). The expression level of PCSK6 is positively correlated with the good prognosis of BC patients (*Yi et al., 2024*). In our risk score model, all six prognostic genes associated with ZHT had HR less than 1, indicating a positive association with better OS in BC.

The GSEA analysis results between the two groups showed that the high-risk group was significantly associated with E2F TARGETS, G2M CHECKPOINT, and KRAS SIGNALING DN pathways. Previous studies have shown that DANCR/miR-34c-5p/E2F transcription factor 1 (E2F1) feedback loop enhances the proliferation, migration and invasion of BC cells (*Yan et al., 2024*). The G2/M checkpoint and E2F targets can also inhibit TNBC

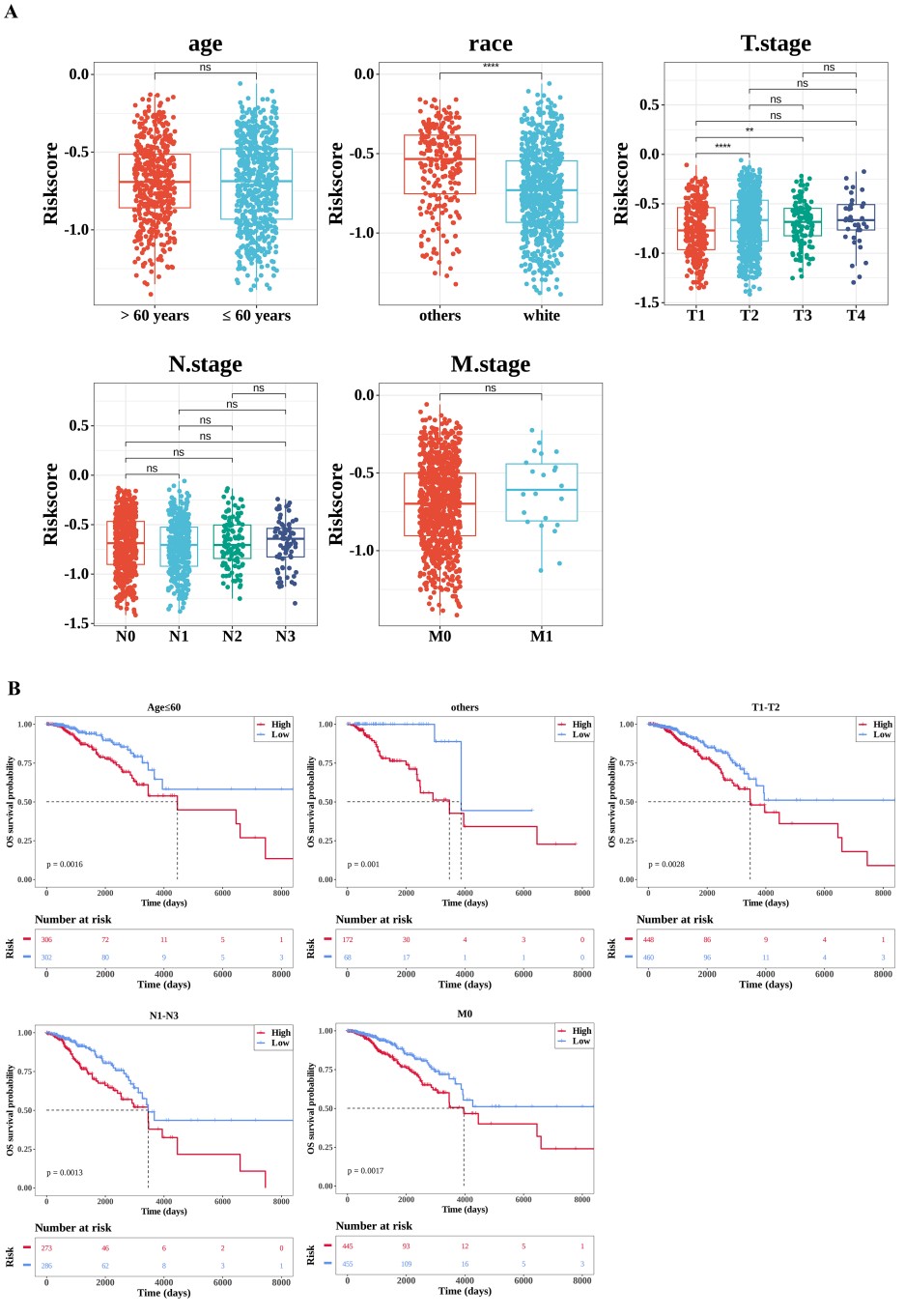

**Figure 5  Correlation analysis of clinical features.** (A) Correlation between the risk score and clinical characteristics. The number of samples in different clinical subgroups was as follows: age >60: $n = 474$, age $\leq60$ $n = 608$; race (others): $n = 240$, race (white) $n = 757$; T1 stage: $n = 281$, T2 stage: $n = 627$, T3 stage: $n = 133$, T4 stage: $n = 38$; N0 stage: $n = 506$, N1 stage: $n = 364$, N2 stage: $n = 120$, N3 stage: $n = 75$; M0 stage: $n = 900$, M1 stage: $n = 22$. (B) Survival analysis of clinical characteristics (age; race (others); T stage; N stage; M stage). The $x$-axis represents time in days, while the $y$-axis shows the OS survival probability. $^{*}P < 0.05$, $^{**}P < 0.01$, $^{***}P < 0.001$, $^{****}P < 0.0001$.

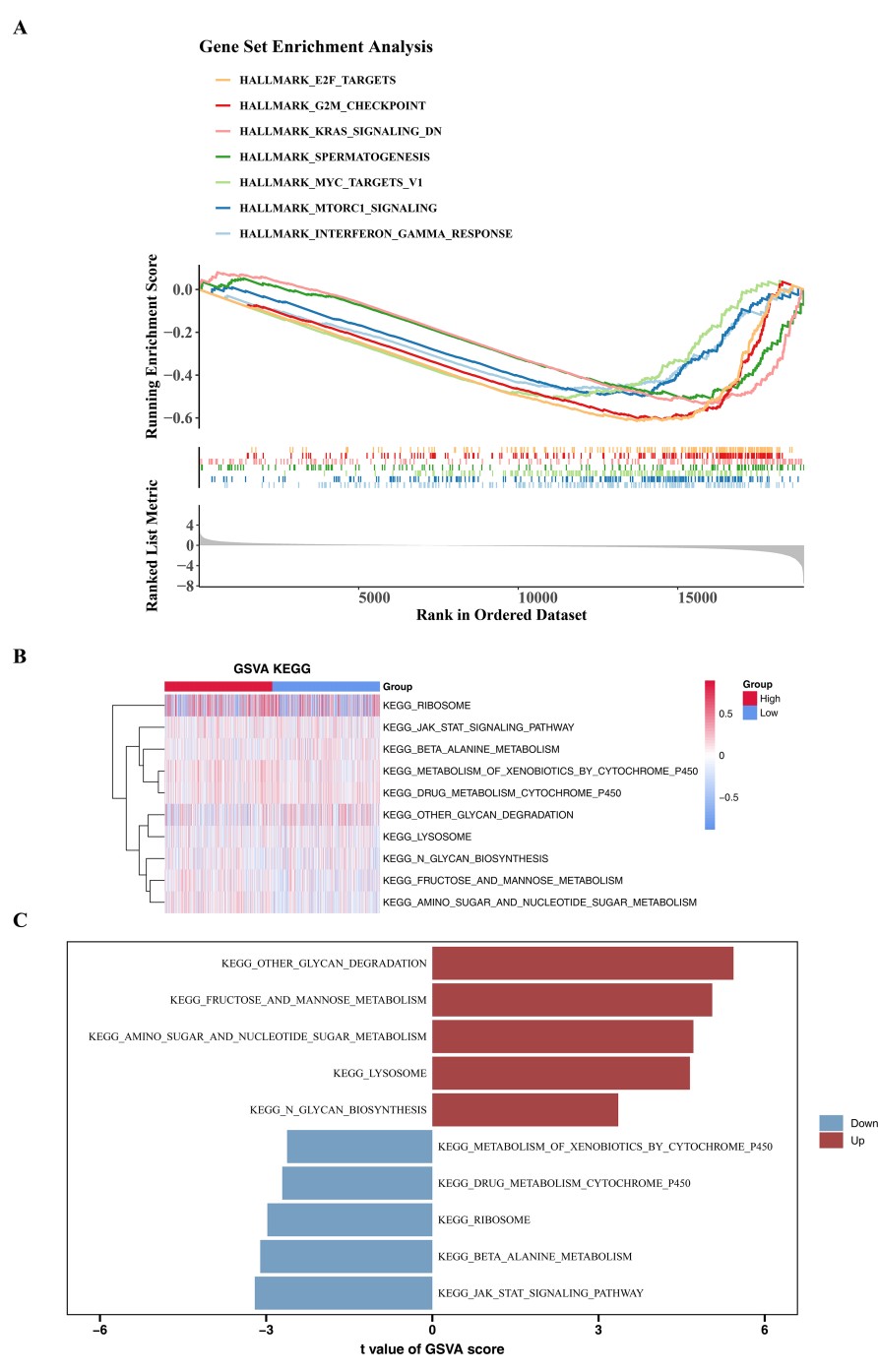

**Figure 6** **GSEA and GSVA analyses.** (A) Enrichment of low and high risk group of Hallmark signaling pathways. The peak value in each line graph represents the enrichment score of this gene set. (B) Heat maps of KEGG genomic GSVA scores in high-risk and low-risk groups, showing the top 10 GSVA analyses. The gradient bar represents the GSVA score, where a redder color indicates a higher GSVA score and a bluer color indicates a lower GSVA score. (C) Up-regulated and down-expressed KEGG pathways in high-risk and low-risk groups. Red represents up-regulated pathways, and blue represents down-regulated pathways.

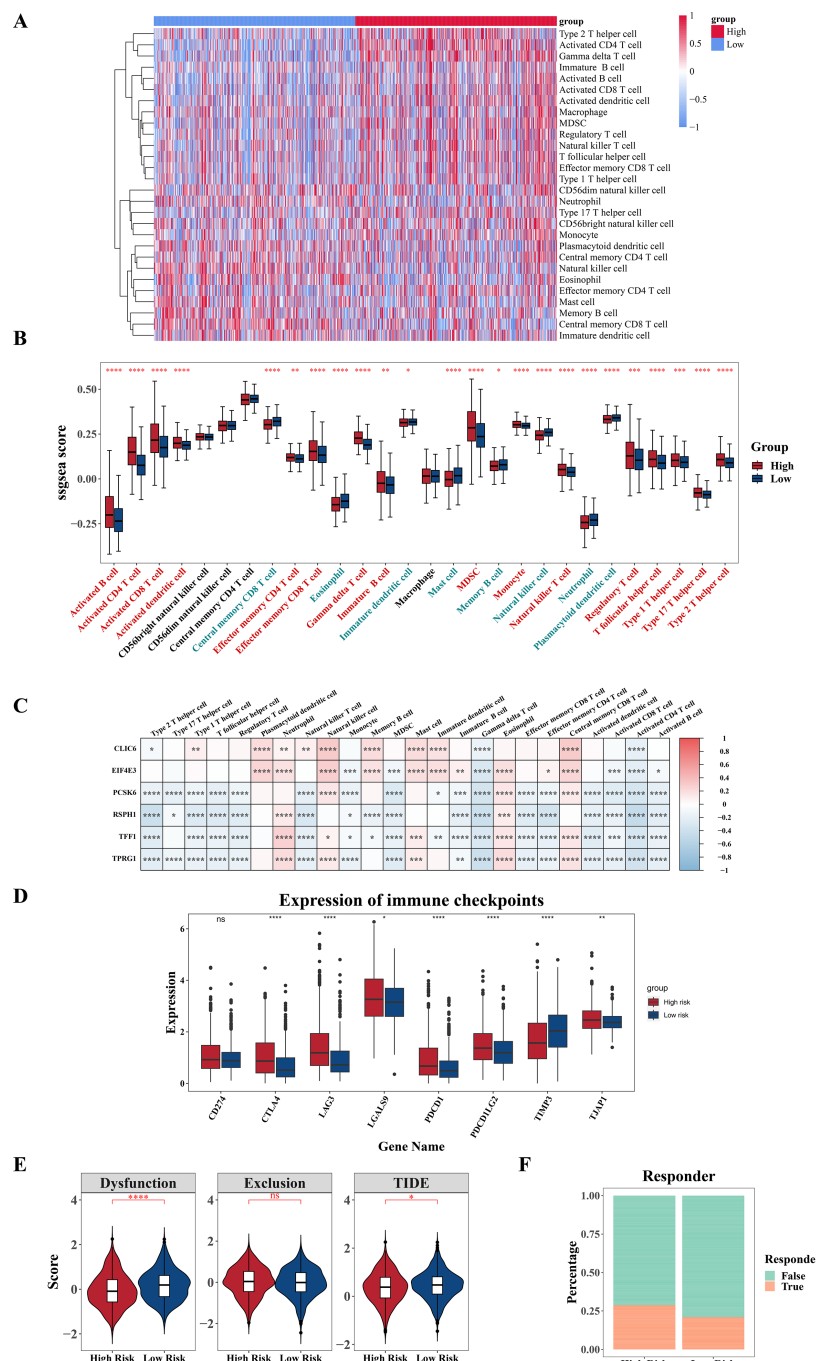

**Figure 7** **Analysis of immune infiltration and immunotherapy.** (A) Heat map shows the enrichment scores of immune infiltrating cells based on the high and low risk groups. The gradient bar represents the levels of the ssGSEA scores of immune cells. The redder the color, the higher the score; the bluer the color, the lower the score. (B) Boxplot shows the abundance of immune infiltrating cells based on the high and low risk groups. (C) Heat map shows the correlation between gene and prognosis of immune cells. Red indicates positive correlation, and blue indicates negative correlation. The darker the color, the stronger

**Figure 7 (…continued)**
the correlation. (D) Boxplot of differences in immune checkpoint molecules between high and low risk groups. (E) Violin plot of TIDE score differences between high and low risk groups. (F) Bar graph of the immune response in the high and low risk groups. *$P < 0.05$, **$P < 0.01$, ***$P < 0.001$, ****$P < 0.0001$.

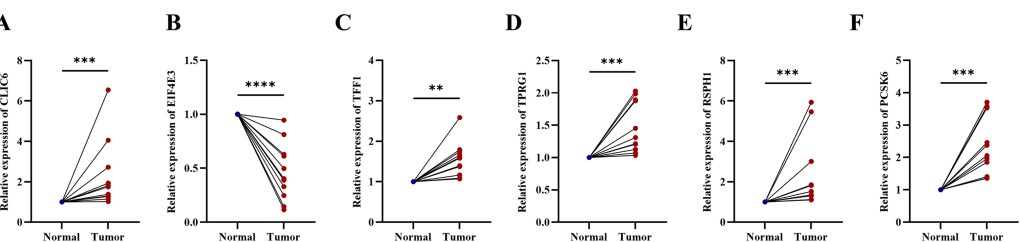

**Figure 8** **Experimental validation of six prognostic genes in tissues of breast cancer patients.** (A–F) Expression of six prognostic genes in 10 paired breast cancer tissues and adjacent normal tissues was evaluated by RT-qPCR. The expression levels of prognostic genes were quantitatively calculated by the $2^{-\Delta\Delta Ct}$ method using $\beta$-actin as the reference gene. *$P < 0.05$, **$P < 0.01$, ***$P < 0.001$, ****$P < 0.0001$.

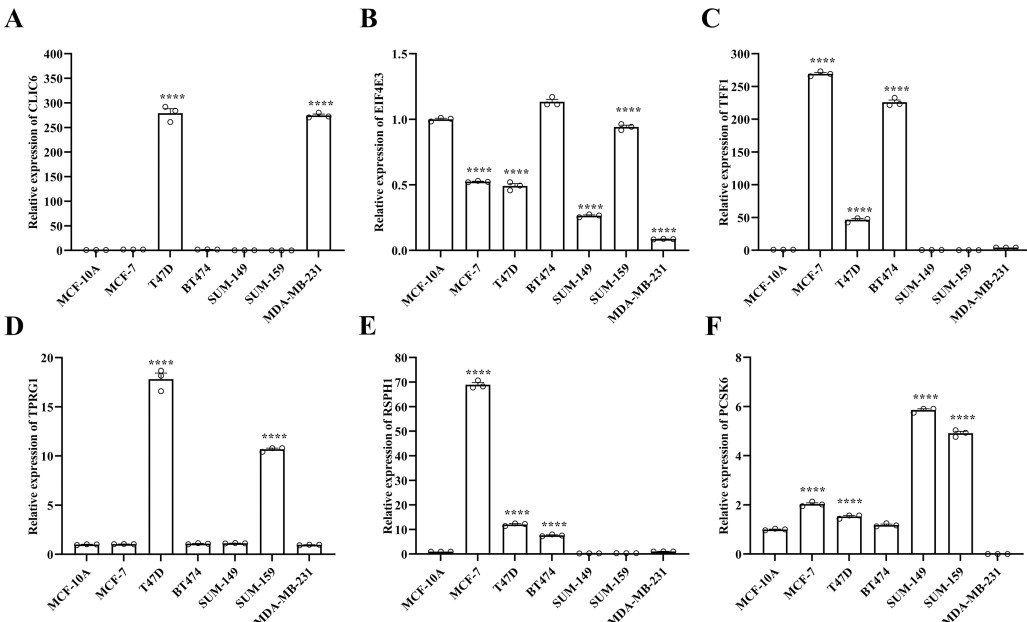

**Figure 9** **Experimental validation of six prognostic genes in cell lines.** (A–F) Expression of six prognostic genes in MCF-10A, T47D, BT474, MDA-MB-231, SUM-149, SUM-159 and MCF-7 cell lines was evaluated by RT-qPCR. The expression levels of prognostic genes were quantitatively calculated by the $2^{-\Delta\Delta Ct}$ method using $\beta$-actin as the reference gene. *$P < 0.05$, **$P < 0.01$, ***$P < 0.001$, ****$P < 0.0001$.

cell growth through endoplasmic reticulum stress-dependent tumor suppressor signaling (*Yang et al., 2024*). Additionally, GSVA also revealed that pathways with differences among two cohorts included OTHER GLYCAN DEGRADATION. Subsequently, we

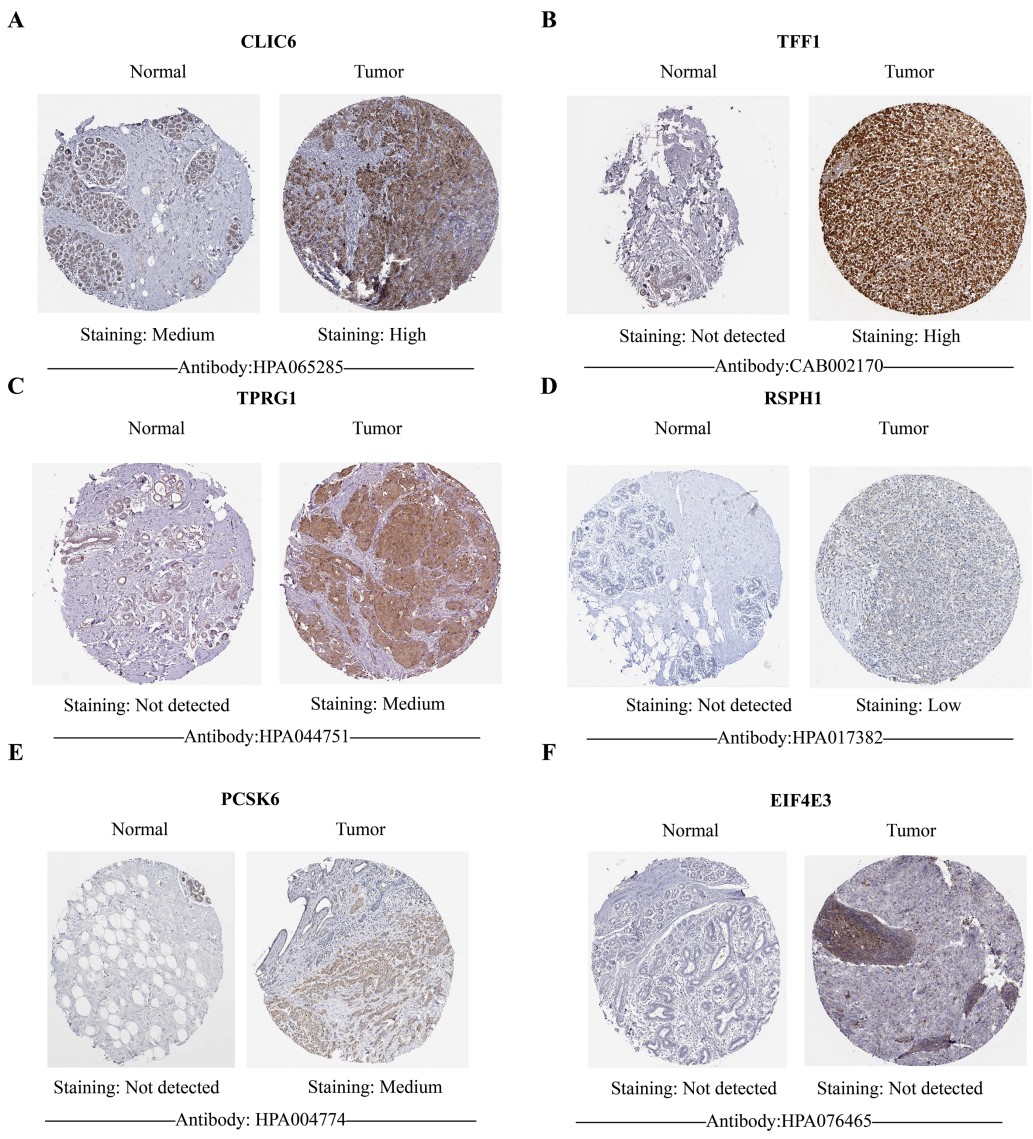

**Figure 10 HPA database verified protein expression levels of prognostic genes.** (A–F) The protein expression of six prognostic genes in normal tissues and breast cancer in the HPA database. Except for EIF4E3, the protein levels of the other five genes showed differential expression between breast cancer tissues and control tissues.

explored the relationship between risk scores and tumor immune microenvironment. We found that eight types of immune infiltrating cells, including NK cells, had higher scores in the low-risk cohort. NK cells possess potent cytotoxicity against infected and cancerous cells and hold promise in the development of new immunotherapies (*Lam & Souza-Fonseca-Guimaraes, 2024*). In addition, NK cells also play an important role in tumor eradication in TME (*Liu et al., 2024a*). Further correlation analysis showed the strongest negative correlation between RSPH1 and activated CD4 T cells. Studies have shown that CD4 T cells mediate direct cytotoxicity to tumor cells through increased production of

interferon gamma (IFN-$\gamma$) and tumor necrosis factor (TNF) (*Subbannayya et al., 2020*). The possible reasons for the influence of tumor microenvironment on tumor biological behavior should be attributed to the comprehensive result of the interaction of multiple immune cells. The underlying mechanism of ZHTGs on BC TME needs to be further investigated. In addition, numerous studies have shown that LAG-3 is the third targeted checkpoint in the clinic (*Ruffo et al., 2019*). Consistent with most previous studies, in our study, the expression levels of many immune checkpoints in high-risk groups were higher, including LAG3, CTLA4, PDCD1, PDCD1LG2, TJAP1, LGALS9, *etc.*

Finally, we verified the expression of six prognostic genes in tumor tissues and normal adjacent tissues, as well as normal breast epithelial cells and BC cell lines by RT-qPCR. This was further confirmed by immunohistochemical results from HPA.

However, this study still has certain limitations. First of all, the clinical information and gene expression data of this prediction model come from a public database, and the validation method is relatively simple, which requires more clinical data to further verify the validity of the model.

Secondly, how prognostic genes regulate zinc homeostasis and the occurrence and development of BC still requires further verification. Therefore, we plan to collect more clinical sample data in the future to verify the effectiveness of the existing model in a broader clinical setting and explore the application potential of the model. In addition, through cell and animal experiments, we will further explore how prognostic genes regulate zinc homeostasis and their roles in the occurrence and development of BC, especially whether these genes function through zinc finger domains or zinc-dependent pathways.

## CONCLUSIONS

We constructed a BC prognosis model related to ZHTGs through a series of bioinformatics methods, analyzed the molecular mechanisms of prognosis genes affecting BC, further explored the role of ZHTGs in the immune microenvironment and immune therapy effects of BC, providing new directions for exploring new immune therapy and targeted treatment strategies. However, the more specific relationship between the prognosis genes and BC requires more experimental data support, and the relevant mechanisms still need further exploration.

### Abbreviations

| | |
|---|---|
| **BC** | Breast cancer |
| **GEO** | Gene Expression Omnibus |
| **GO** | Gene Ontology |
| **GS** | Gene Significance |
| **GSEA** | Gene Set Enrichment Analysis |
| **GSVA** | Gene Set Variation Analysis |
| **HPA** | Human Protein Atlas |
| **HR** | Hazard Ratio |
| **KEGG** | Kyoto Encyclopedia of Genes and Genomes |

| K-M | Kaplan–Meier |
|---|---|
| LASSO | Least Absolute Shrinkage and Selection Operator |
| log2FC | log2Fold Change |
| MM | Module membership |
| MSigDB | Molecular Signatures Database |
| OS | Overall Survival |
| PH | Proportional Hazards |
| RT-qPCR | Real-Time Quantitative PCR |
| ssGSEA | single-sample Gene Set Enrichment Analysis |
| TCGA | The Cancer Genome Atlas |
| TIDE | Tumor Immune Dysfunction and Exclusion |
| WGCNA | Weighted gene co-expression network analysis |
| ZHTGs | Zinc homeostasis and Zinc transporter-related genes |

## ACKNOWLEDGEMENTS

We are very grateful for the data provided by databases such as TCGA, GEO, and HPA *etc.* We thank the Life Science and Medical Analysis and Testing Laboratory of the Air Force Medical University for allowing us to conduct all experimental research described in this article on their laboratory platform.

### Funding

This work was supported by the Cultivation Boost Project of Xijing Hospital (XJZT24LY09) and Shaanxi Province Natural Science Basic Research Program Key Project (2021JZ-29, 2023-JC-QN-0965). The funders had no role in study design, data collection and analysis, decision to publish, or preparation of the manuscript.

### Grant Disclosures

The following grant information was disclosed by the authors:
Cultivation Boost Project of Xijing Hospital: XJZT24LY09.
Shaanxi Province Natural Science Basic Research Program Key Project: 2021JZ-29, 2023-JC-QN-0965.

### Competing Interests

The authors declare there are no competing interests.

### Author Contributions

- Mengxuan Li conceived and designed the experiments, performed the experiments, prepared figures and/or tables, and approved the final draft.
- Haoyi Zi conceived and designed the experiments, performed the experiments, prepared figures and/or tables, and approved the final draft.
- Jiajun Ding conceived and designed the experiments, performed the experiments, prepared figures and/or tables, and approved the final draft.

- Shuai Wang performed the experiments, prepared figures and/or tables, and approved the final draft.
- Yujie Bai analyzed the data, prepared figures and/or tables, and approved the final draft.
- Jianing Sun analyzed the data, prepared figures and/or tables, and approved the final draft.
- Cong Fan analyzed the data, prepared figures and/or tables, and approved the final draft.
- He Chen analyzed the data, prepared figures and/or tables, and approved the final draft.
- Ting Wang conceived and designed the experiments, authored or reviewed drafts of the article, and approved the final draft.

## Human Ethics

The following information was supplied relating to ethical approvals (i.e., approving body and any reference numbers):

The patients involved in the database have obtained ethical approval. Approval was granted by the Ethics Committee of Xijing Hospital of Air Force Medical University (KY20232266-C-1). Informed consent was obtained from all individual participants included in the study.

## Data Availability

The raw measurements are available in the Supplemental Files.

The TCGA-BRCA data is available at GDC Data Portal: Available at https://portal.gdc.cancer.gov/projects/TCGA-BRCA.

The microarray data is available at GEO: GSE20685.

The CLIC6 BREAST INVASIVE CARCINOMA data is available at: Available at https://www.proteinatlas.org/ENSG00000159212-CLIC6/cancer/breast+cancer#img.

The TFF1 BREAST INVASIVE CARCINOMA data is available at: Available at https://www.proteinatlas.org/ENSG00000160182-TFF1/cancer/breast+cancer#img.

The TPRG1 BREAST INVASIVE CARCINOMA data is available at: Available at https://www.proteinatlas.org/ENSG00000188001-TPRG1/cancer/breast+cancer#img.

The RSPH1 BREAST INVASIVE CARCINOMA data is available at: Available at https://www.proteinatlas.org/ENSG00000160188-RSPH1/cancer/breast+cancer#img.

The PCSK6 BREAST INVASIVE CARCINOMA data is available at: Available at https://www.proteinatlas.org/ENSG00000140479-PCSK6/cancer/breast+cancer#img.

The EIF4E3 BREAST INVASIVE CARCINOMA (TCGA) data is available at: Available at https://www.proteinatlas.org/ENSG00000163412-EIF4E3/cancer/breast+cancer#img.

## Supplemental Information

Supplemental information for this article can be found online at http://dx.doi.org/10.7717/peerj.20031#supplemental-information.

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
