# Peer review of "Identification and verification of prognostic genes related to zinc homeostasis and zinc transport in breast cancer"

_PeerJ, doi:10.7717/peerj.20031_

## Round 0.1 · original submission · Major Revisions

Please carefully address the comments of both reviewers and amend the manuscript accordingly.

Reviewer 1 ·

Basic reporting

The manuscript addresses a well-defined and clinically relevant research question. Building on prior evidence that zinc transporter-related genes are implicated in cancer development, the authors propose that constructing a prognostic model based on Zinc transporter-related genes could improve risk stratification in breast cancer patients. To this end, they employed a combination of bioinformatics approaches using both publicly available datasets and patient samples from the Xijing Hospital, ultimately identifying six genes described as “prognosis genes related to Zinc transporter-related genes in breast cancer, providing a reference for the prognosis and personalized treatment of breast cancer.”

However, the current version would greatly benefit from clearer and more didactic reporting. While the overall structure is consistent with standard scientific articles, there are several areas where clarity, precision, and accessibility could be improved, particularly in the use of acronyms, methodological explanations, and figure presentation. At times, the writing assumes a high level of familiarity with specific tools and metrics, which may limit accessibility for a broader readership. Enhancing the descriptions of analytical approaches (e.g., WGCNA, TIDE), defining key metrics, and improving figure resolution and legends would significantly increase the manuscript’s clarity and scientific value. Additionally, attention to minor typographical errors and ambiguous phrasing would further enhance readability. Overall, I believe the manuscript has strong potential, and with more attention to detail and accessibility, it could make a valuable contribution to the field.

- Ambiguity:
a) Clarification of WGCNA
The manuscript would benefit from a brief explanation of what WGCNA stands for, what the approach allows, and why it is particularly suited for identifying prognostic genes in breast cancer. While the exhaustive procedure is described in the “Materials and Methods” section, its rationale and relevance are not clearly introduced in the main text. The current description is quite limited:
(lines 110–112)“Next, the WGCNA package (v 1.70.3) (Langfelder & Horvath 2008) was employed to construct a co-expression network using ZHTGs scores as traits, aiming to identify module genes most correlated with the traits.”
In contrast, the cited source provides a more informative overview of WGCNA’s utility and perhaps the authors could adopt a similar approach to add a short explanatory sentence in the Results or Introduction, such as around line 233, as it would help readers unfamiliar with the method better understand its purpose and importance in the study:
Langfelder & Horvath 2008: “Weighted correlation network analysis (WGCNA) can be used for finding clusters (modules) of highly correlated genes. […]. The WGCNA R software package is a comprehensive collection of R functions for performing various aspects of weighted correlation network analysis.”

b) Clarification of TIDE and dysfunction scores
At line 306, the manuscript references both the “TIDE score” and “Dysfunction score”, but no explanation is provided regarding what these metrics represent. This may limit accessibility for readers who are not already familiar with these computational tools. More broadly, the manuscript would benefit from clearer descriptions of the key methods and metrics used throughout the study. The analytical approach and results are compelling, but I would encourage the authors to guide the reader more clearly through the data interpretation process, particularly given that this submission is intended for “PeerJ – the multidisciplinary journal of Life, Medical & Environmental Sciences”, and not for “PeerJ Computer Science”.

c) Acronym usage – BRCA vs. BC
I would recommend reconsidering the use of the acronym “BRCA” throughout the manuscript. It may cause confusion, particularly in a context where the BRCA1 and BRCA2 genes are also highly relevant. Using “BC,” which is more widely accepted and unambiguous in the field, could help prevent potential misinterpretation.

d) Clarification regarding the MCF-10A cell line
At line 314, the authors state:
“In addition, we also examined the expression of prognostic genes in the BRCA cell lines, MCF-10A, T47D, BT474, MDA-MB-231, SUM-149, SUM-159, and MCF-7.”
However, MCF-10A is a non-tumorigenic epithelial cell line and is typically used as a control rather than a breast cancer cell line. This appears to be a minor typo or wording issue, as the authors correctly used this cell line as a control in their qPCR experiments (Figure 9).

e) Sentences requiring clarification
Several sentences throughout the manuscript are unclear and would benefit from revision to avoid potential misunderstanding by the readers:

- Lines 79–81:
“Therefore, this study is based on the transcriptome data of BRCA in The Cancer Genome Atlas (TCGA) and Gene Expression Omnibus (GEO) databases and the gene set of ZHTGs.”
This sentence is incomplete, and the absence of a verb makes it difficult to interpret.

- Line 224:
“In the TCGA-BRCA dataset, there were 4,948 DEGs were discovered […]”
The sentence appears to contain a duplication (“there were… were discovered”) and should be revised for clarity and grammar.

- Line 234:
“With a B value of 6, an R2 approaching 0.85, and a connectivity close to 0 were achieved (Fig.1E).”
This sentence is syntactically ambiguous. It is unclear whether “a B value of 6” is one of the outcomes or whether it is the condition under which the other two values (R² and connectivity) were achieved. The comma placement and parallel structure suggest a list of three outcomes, but the intended meaning seems to be: by setting B to 6, an R² approaching 0.85 and a connectivity close to 0 were obtained.
To improve clarity, the authors might consider rephrasing the sentence. For example:
“By setting the B value to 6, we achieved an R² approaching 0.85 and a connectivity close to 0 (Fig.1E).” This would more clearly convey the relationship between the variables and the result.

- Line 278:
“Subsequently, additional comparisons were made regarding the disparities in risk scores across various clinical feature sub-cohorts, revealing significant survival disparities among two cohorts in sub-cohorts characterized by Agef60, race (others), T.Stage 1-2, N.stage1-3, and M.stage0.”
This sentence is overly long and complex. Breaking it into shorter, clearer statements would improve readability.

- Line 316:
“Compared with the MCF10A cell line, the gene expression level in most cell lines was basically consistent with the tissue verification results.”
The term “basically” is informal and vague in a scientific context. Consider replacing it with “overall” or “generally.”

f) Statistical interpretation of gene expression differences
At lines 319–320, the authors state:
“CLIC6, TFF1, TPRG1, RSPH1, and PCSK6 were expressed at significantly different levels between BRCA and normal breast tissues. There was no significant difference in the expression of EIF4E3 (Fig. 10A–F).”
However, no statistical test appears to be reported in the figure or the main text to support these statements. I would recommend rephrasing to avoid implying statistical significance and perhaps use “drastically” or “notably”.

- Literature references, sufficient field background/context provided:
The authors provide a clear and appropriate background on breast cancer and zinc metabolism in the “Introduction”, as well as relevant references to the databases, WGCNA, and risk model in the “Methods” section. The “Discussion” section also effectively cites literature related to the six prognostic genes identified. Overall, the manuscript is well-anchored in the existing literature and provides sufficient context to support the study’s rationale and findings.

- Professional article structure, figures, and tables. Raw data shared
a) The figure legends would benefit from greater detail and consistency. Figures 1 and 2 serve as good examples, as their legends are self-explanatory and clearly define key metrics (e.g., z-score). In contrast, legends for Figures 4, 5, 6, 8, 9, and 10 could be improved by including essential methodological information. For instance, the legend for Figure 8 states:

“Experimental validation of 6 prognostic genes in tissues. (A–F) Expression of 6 prognostic genes in 10 paired breast cancer tissues and adjacent normal tissues was evaluated by RT-qPCR * P <0.05, ** P <0.01, *** P <0.001, **** P<0.0001.”

However, it does not clarify the source of the tissues used—whether they were obtained from a clinical trial, a database, or another source (although detailed in the “Methods” section). Additionally, the legend could briefly indicate that the authors used the 2^delta-delta-Ct method to calculate relative gene expression (although described in the “Methods” section). It would also be helpful to specify which housekeeping gene(s) were used for normalization, either in the legend or directly on the plots. Additionally, this information is missing from both the “Methods” section and the “supplementary file Table_S2.xlsx”, which lists primer sequences but does not include primers for a control gene.

b) Several figures are presented at low resolution, which affects readability and interpretation. This is particularly noticeable in: Figure 1A, 1C (table under the KP curve), 1G; Figure 2B (table and gradient bar);
Figure 3B, 3E (tables under the KP curves); Figure 5B (table under the KP curve); and Figure 7C (cell type annotation)
The manuscript would benefit from higher-resolution versions of these figures. Additionally, reducing whitespace between panels—or selectively removing less critical panels—could help allocate more space to key visual elements and improve overall clarity and visual accessibility.

c) The supplementary file “Raw data of RT-qPCR” does not appear to contain raw data in the strict sense, as the values have already been processed. Sharing truly raw values would improve transparency and reproducibility.

d) The x-axis label in Figure 3A contains a typographical error:
“Patients (increasing risk score)”
This should be corrected to “Patients (increasing risk score).”

e) In the legend of Figure 4, panel C is referenced in the figure but not cited or described in the legend. This should be addressed for better comprehension.

f) The legend of Figure 5 should state the number of samples per group (as the groups appear to be very asymmetrical) (e.g., T-stage T4 or M-stage M1 have notably fewer points than other groups).

g) In Figure 6B, it is unclear what the gradient bar on the right (–0.5 to 0.5) represents. Is it enrichment, a p-value, or another metric? This should be explained clearly in the figure legend.

h) Consistency and clarity of color schemes
The heatmaps in the manuscript (e.g., Figures 3 and 7) use a red–blue color gradient, which is standard and effective for representing expression levels. However, the colors used to indicate high- and low-risk groups vary across figures, alternating between dark blue & light blue (high risk) and dark red & pink (low risk). This inconsistency can create confusion, particularly in heatmaps like Figure 7, where the group colors can conflict visually with the heatmap gradient.
I would strongly encourage the authors to standardize their color scheme throughout the manuscript, either by selecting a distinct and consistent palette for group classification or by adjusting the heatmap colormap to avoid overlap. Ensuring color consistency across the risk group annotations and also the Kaplan–Meier curves would enhance visual clarity and reader comprehension.

i) The section titled “Clinical feature was associated with risk score and survival rate” (lines 275–282) would benefit from additional detail and clarity. I would strongly encourage the authors to follow the model of the subsequent section (“Patients in different risk cohorts had different effects on the immune response”, lines 290–309), which effectively walks the reader through the analysis performed and presents the findings in a didactic and accessible way.

j) I would like to highlight that the distinction between Supplementary Figures 3 and 4 is not immediately clear, aside from one showing positive correlation and the other negative correlation. Adding legends to these figures and further elaborating on their relevance in the main text of the Results section would improve clarity and interpretation.

- Self-contained with relevant results to hypotheses
The manuscript is self-contained and integrates results from both publicly available databases and the authors’ own experimental work, including cell line data and samples from the Xijing Hospital. The authors present their findings transparently and avoid overstating their conclusions. They clearly acknowledge the need for further validation and thoughtfully discuss the limitations of their study, which adds to the credibility and scientific integrity of the work.

Experimental design

This study aligns well with the Aims and Scope of PeerJ within the fields of Medical and Health Sciences. In the Introduction section, the authors contextualize the relevance of breast cancer research and state:

“It is important to further explore the relevant characteristics of BRCA and potential new therapeutic targets.”
They then introduce the role of zinc metabolism in breast cancer and conclude with a clear articulation of their research objective:
“By using bioinformatics methods, a BRCA prognostic model related to ZHTGs was constructed to explore the BRCA immune microenvironment and immune therapy effects, providing a new direction for discovering new immune therapy and targeted therapy strategies.”

This objective is well-aligned with the analyses carried out using publicly available datasets such as The Cancer Genome Atlas (TCGA) and Gene Expression Omnibus (GEO).

While the cost-saving aspect of using public datasets is well known, an additional strength lies in the fact that these datasets were generated under strict ethical and scientific standards. Additionally, the authors further support reproducibility by using a widely recognized R package (WGCNA), cited more than twelve thousand times, and by sharing their code in the supplementary materials (Source code, Code.zip). They also indicate compliance with the Minimum Information for Publication of Quantitative Real-Time PCR Experiments (MIQE) guidelines, which adds to the methodological rigor of their experimental validation.

Overall, the experimental design appears robust and well-documented. The combination of trusted computational tools, transparent code sharing, and detailed methodological reporting ensures that the work is both reproducible and scientifically sound.

Validity of the findings

The findings presented in the manuscript appear to be statistically grounded and carefully interpreted. The underlying data are derived from trusted public repositories (TCGA, GEO), and the authors provide code and methodological details that support reproducibility. As noted earlier, while some figure legends and statistical statements would benefit from clarification, the overall data handling and analysis appear sound.

Importantly, the authors mostly avoid overstating their conclusions. They clearly link their findings back to the original research question and acknowledge the need for further work to validate their observations. Their conclusions remain within the scope of the results presented and are framed in a measured and scientifically appropriate manner.

I would nonetheless like to bring the authors’ attention to a few points that may benefit from slight rewording to prevent potential misinterpretation by readers:

- line 257-258: “It was noticed that as the risk scores rose in the BRCA specimens, the number of deaths significantly increased”
It is unclear whether statistical significance is actually demonstrated in the corresponding risk score plot. Clarifying whether a formal test was performed would help support this statement. If no statistical test was performed, the authors might reconsider using the term “significantly” and alternatively use “notably” or any other term conveying the same idea, without implying statistical validation.

- line 260-262: “Furthermore, ROC curves were plotted at 1, 3, and 5 years as survival time nodes, demonstrating that the AUC values at all three time points were higher than 0.6, suggesting robust forecasting ability of the model (Fig.3C).”
Since an AUC of 0.5 reflects a model performing no better than random chance, the use of “robust” to describe an AUC of approximately 0.65 may overstate the model’s predictive power. I would suggest softening this language to maintain alignment with the data.
Conversely, the validation dataset appears to show a clearer separation between survival groups than the training cohort. This point strengthens the manuscript and could be more clearly emphasized in the results and conclusions.

Reviewer 2 ·

Basic reporting

In this paper, the authors have investigated the prognostic role of Zinc homeostasis and Zinc transporter-related genes (ZHTGs) in BRCA and their impact on the tumour microenvironment.
They report the six prognostic genes related to ZHT in BRCA.
While the abstract was attractive and the question of interest, many concerns exist about the study's purpose, title, and implications.

The study mentions zinc homeostasis, but no information is given in the introduction about zinc transporter proteins, which are key elements of this homeostasis. This is probably because 14 zinc channel proteins either import or export functions, were not significantly observed in the analyses.

However, SLC30A5 (Znt5) channel protein was found to be expressed in the high-risk group (Figure 1), but these zinc channel proteins were not mentioned again.

In a study on zinc homeostasis, the omission of channel proteins was considered a major deficiency.

The introduction is insufficient to highlight the findings

Experimental design

The study is mainly based on bioinformatics analyses of existing databases and linking the findings with clinical features.

- The relationship of these genes with zinc is not mentioned in the text. At this point, the term zinc-related is not understood (e.g., these proteins have zinc finger domains or act as zinc chelators)

-How were the selected cell lines chosen? e.g., poor or well differentiated is important to show prognosis, but such a comparison has never been made in any study.

- After discrimination of the differentiation level of BRCA cell lines, the intracellular zinc level should also be demonstrated.

Validity of the findings

In this study, bioinformatics analyses were performed from quite different perspectives, and genes and pathways that may be effective in prognosis were identified. However, the relationship of these genes with zinc homeostasis has not been explained and is unclear.
Therefore, we can say that the title of the study does not indicate the content.

---

## Round 0.2 · Minor Revisions

Please address remaining concerns of the reviewer and amend manuscript accordingly.

Reviewer 1 ·

Basic reporting

The manuscript addresses a well-defined and clinically relevant research question. Building on prior evidence that zinc transporter-related genes are implicated in cancer development, the authors propose that constructing a prognostic model based on Zinc transporter-related genes could improve risk stratification in breast cancer patients. To this end, they employed a combination of bioinformatics approaches using both publicly available datasets and patient samples from the Xijing Hospital, ultimately identifying six genes described as “prognosis genes related to Zinc transporter-related genes in breast cancer, providing a reference for the prognosis and personalized treatment of breast cancer.”

The authors clarified most of the points that were debatable in the first reviewing round and the author’s manuscript had greatly improved. It is now more transparent and didactic. The figures offer a better resolution, with more descriptive legends, and acronyms have been clarified. A few minor typographical errors remain and should be corrected by the authors. Overall, these additions make the manuscript more robust and will constitute a valuable contribution to the field.

- Line 103-104: The authors did format the reference, and the publication they cite is referred as “PMID: 19114008”
- Line 301-302: “To explore the significantly enriched pathways between the high- and low-risk groups, was performed.”
The sentence misses the type of analyses that were performed (“GSEA and GSVA”, as stated in the previous version of the manuscript).

Experimental design

This study aligns well with the Aims and Scope of PeerJ within the fields of Medical and Health Sciences. In the Introduction section, the authors contextualize the relevance of breast cancer research and state:
“It is important to further explore the relevant characteristics of BRCA and potential new therapeutic targets.”

They then introduce the role of zinc metabolism in breast cancer and conclude with a clear articulation of their research objective:
“By using bioinformatics methods, a BRCA prognostic model related to ZHTGs was constructed to explore the BRCA immune microenvironment and immune therapy effects, providing a new direction for discovering new immune therapy and targeted therapy strategies.”

This objective is well-aligned with the analyses carried out using publicly available datasets such as The Cancer Genome Atlas (TCGA) and Gene Expression Omnibus (GEO).

While the cost-saving aspect of using public datasets is well known, an additional strength lies in the fact that these datasets were generated under strict ethical and scientific standards. Additionally, the authors further support reproducibility by using a widely recognized R package (WGCNA), cited more than twelve thousand times, and by sharing their code in the supplementary materials (Source code, Code.zip). They also indicate compliance with the Minimum Information for Publication of Quantitative Real-Time PCR Experiments (MIQE) guidelines, which adds to the methodological rigor of their experimental validation.

Overall, the experimental design appears robust and well-documented. The combination of trusted computational tools, transparent code sharing, and detailed methodological reporting ensures that the work is both reproducible and scientifically sound.

Validity of the findings

The findings presented in the manuscript appear to be statistically grounded and carefully interpreted. The underlying data are derived from trusted public repositories (TCGA, GEO), and the authors provide code and methodological details that support reproducibility. As noted earlier, while some figure legends and statistical statements would benefit from clarification, the overall data handling and analysis appear sound.

Importantly, the authors mostly avoid overstating their conclusions. They clearly link their findings back to the original research question and acknowledge the need for further work to validate their observations. Their conclusions remain within the scope of the results presented and are framed in a measured and scientifically appropriate manner.

---

## Round 0.3 · accepted · Accept

All remaining issues were addressed and the revised manuscript is acceptable now.